# Suppression of pathogens in properly refrigerated raw milk

**M. E. Coleman**[1]\*, **T. P. Oscar**[2], **T. L. Negley**[3], **M. M. Stephenson**[4]

**1** Coleman Scientific Consulting, Groton, NY, United States of America, **2** USDA, Agricultural Research Service, Princess Ann, MD, United States of America, **3** TIG Environmental, Syracuse, NY, United States of America, **4** Advancement and External Affairs, Syracuse University, Syracuse, NY, United States of America

\* peg@colemanscientific.org

## Abstract

Conflicting claims exist regarding pathogen growth in raw milk. A small pilot study was designed to provide definitive data on trends for pathogen growth and decline in raw bovine milk hygienically produced for direct human consumption. An independent laboratory conducted the study, monitoring growth and decline of pathogens inoculated into raw milk. Raw milk samples were inoculated with foodborne pathogens (*Campylobacter*, *E. coli* O157:H7, *Listeria monocytogenes*, or *Salmonella*) at lower (<162 colony forming units (CFU) per mL) and higher levels (<8,300 CFU/mL). Samples were stored at 4.4°C and quantified over time after inoculation (days 0, 3, 6, 9, 12, and 14) by standard culture-based methods. Statistical analysis of trends using the Mann-Kendall Trend Test and Analysis of Variance were conducted for 48 time series observations. Evidence of pathogen growth was documented for *L. monocytogenes* in 8 of 12 replicates (P = 0.001 to P = 0.028). Analysis of variance confirmed significant increases for *L. monocytogenes* at both initial levels in week 2. No evidence of growth was documented over 14 days for the three pathogens predominantly associated with raw milk outbreaks in the US (*Campylobacter*, *E. coli* O157:H7, and *Salmonella*). Further research is needed to characterize parameters for pathogen growth and decline to support re-assessment of risks that were based on incorrect assumptions about interactions of pathogens with the raw milk microbiota.

## Introduction

Evidence exists that humans have consumed ruminant milks for millennia [1] well before milk pasteurization became common in the 20th century. In recent decades, unpasteurized (raw) milk has been legally available for direct human consumption in most US states [2] and in many countries around the world [3]. Evidence from Organizations including the Raw Milk Institute (RAWMI, Fresno, CA USA, https://www.rawmilkinstitute.org/) and the Raw Milk Producers Association (RMPA, Suffolk, UK, https://www.rawmilkproducers.co.uk/) advocate well-documented risk management procedures, including farmer training and mentoring, use of food safety plans similar to 'farm-to-table' or 'grass-to-glass' Hazard Analysis and Critical Control Points (HACCP) procedures, and stringent routine testing for bacterial indicators of

**Funding:** The Raw Milk Institute, (Fresno, CA, 93706, USA) provided funding for both the pilot

study conducted by an independent certified laboratory and the statistical analysis of its results documented herein. The funders had no role in study design, data collection and analysis, decision to publish, or preparation of the manuscript.

**Competing interests:** M.E.C. serves as an unpaid advisor on the Raw Milk Institute advisory board and has provided paid consulting services to the Raw Milk Institute and expert testimony in several court cases regarding the microbial ecology and assessment of benefits and risks for microbial pathogens in raw milk and other foods and feeds. This does not alter our adherence to PLOS ONE policies on sharing data and materials. There are no restrictions on use of the minimal dataset provided as Appendix 3 in the Supporting Information section. Other authors have no competing interests to declare.

potential contamination. Some US states license dairy farms and periodically monitor microbial indicators and pathogens in raw milk produced for direct human consumption, and one farm currently applies test-and-hold procedures for major bacterial pathogens before each lot of raw milk is bottled for consumers in California retail markets [4].

Carefully produced hygienic raw milk for direct human consumption has become associated with health and rarely with foodborne disease outbreaks as documented in recent studies [2, 3]. Dietert and colleagues [2] cited extensive evidence from monitoring programs of six countries reporting rare pathogen detection (<0.01%) in raw milk produced for human consumption, as distinguished from higher rates reported for pre-pasteurized milk of undetermined quality [3]. The Dietert study also compiled evidence of health benefits for raw milk consumers, and no outbreaks of illness from 2018 to 2020 in CA when more than 1,352,000 gallons of fluid raw milk was provided in the CA retail market, consistent with a risk of illness less than 1 in over 20 million 250-mL servings for retail raw milk consumers. Notably, the US Food and Drug Administration (FDA) and the Food Safety and Inspection Service (FSIS) jointly reported that both raw and pasteurized milks were high risk foods for severe listeriosis [5], and a recent systematic review reported that severe listeriosis risks were significantly higher for pasteurized than raw milks [6].

## Literature on predictive microbiology of raw and pasteurized milks

Raw milk producers, regulators, and consumers need reliable and statistically rigorous data to inform their decisions about the safety of raw milk for direct human consumption. Although researchers have understood for decades that rates of growth of pathogens inoculated into raw milk were slower than rates measured in pasteurized milk treated under the same conditions [7–11], this knowledge has not yet been integrated into risk analysis processes (risk assessment, risk communication, risk management) or policies about managing raw and pasteurized milks. Suppression of pathogen growth is attributed to bioactive components of milk, including competition with the microbes naturally present in raw milks, the milk microbiota. Strong evidence from both traditional culturing and culture-independent methodology has accumulated in this decade characterizing the natural milk microbiota [12–14] and its crucial role in balancing benefits and risks for human health [2, 15]. Recent studies [16–18] consider contributions of raw and pasteurized milks to the current epidemic of allergic, inflammatory, and non-communicable diseases that merit simultaneous considerations of benefits and risks attributable to both infectious and non-communicable diseases for assessing human health and well-being. Unfortunately, misinformation about raw milk and the interactions of its natural microbiota with potential pathogens and host cells abounds, even in the peer reviewed literature and government documents.

The raw milk microbiota of mammals commonly includes lactic acid bacteria or LAB [12, 14]. Many LAB strains can outcompete pathogens by competing for nutrients as well as by active antagonism via bacteriocins and other microbial metabolites [19, 20]. Many of the diverse microbes classified as LABs are common members of the milk microbiota [12, 14, 20], including many Generally Recognized as Safe (GRAS) or Qualified Presumption of Safety (QPS) [21, 22] with a safe history of use as probiotics (e.g., *Bifidobacterium*, *Enterococcus*, *Lactobacillus*, *Lactococcus*, *Leuconostoc*, *Pediococcus*, and *Streptococcus*) that also appear to contribute to human and animal health [12, 14, 23, 24]. Further, raw milk including the natural microbiota significantly suppressed adherence, invasion, and proliferation of a high dose of *L. monocytogenes* to human intestinal line cells versus administration of the pathogen in pasteurized milk or buffer [25]. Greater understanding of the interactions of the raw milk microbiota with pathogens, both in our refrigerators and in the human gut, as well as their mechanisms of

protection, is needed to appropriately model benefits and risks that raw milk microbes pose in complex ecosystems.

Seven studies were identified in our literature searches (S1 Table) that reported data on growth and survival of pathogens inoculated into raw milk and incubated at refrigeration temperatures. Of these, three studies [7, 8, 26] monitored pathogen growth in both raw and pasteurized milks. All three studies documented either no pathogen growth at 4–5°C or slower growth in raw milk including the natural microbiota compared to pasteurized milk with greatly diminished microbial competitors.

Despite documentation in the published literature of higher pathogen growth rates in pasteurized milks, FDA/FSIS [5] assumed in its quantitative microbial risk assessment (QMRA) for listeriosis in Ready-to-Eat foods that growth of the pathogen *L. monocytogenes* was equivalent in raw and pasteurized milks. FDA/FSIS [5] reported an 'average' growth rate of 0.257 hr$^{-1}$ for milk in the body of the QMRA report, and documented pooling of the data of [7], 0.085 for raw milk and 0.407 for pasteurized milk adjusted to 5°C, in Appendix 8 of the QMRA report [5].

Early predictive microbiology studies demonstrated the importance of time, temperature and the initial inoculation density of pathogens inoculated into sterile culture broth as the boundary for the growth/no-growth interface for the pathogen *E. coli* O157:H7 (~10°C) was approached [27, 28]. Clear dependencies were documented for initial pathogen density and temperature in broth culture studies and simulations of growth in non-sterile foods [28, 29]. However, non-sterile foods including raw milk are expected to impose additional limitations on pathogen growth and acceleration of pathogen decline due to the presence of a natural microbiota [30] and other biologically active components including enzymes and bacteriocins that suppress pathogens [31].

To document the mathematical relationships for pathogen growth and decline in raw milk for assessing and re-assessing microbial risks, a study design is needed that takes into account available knowledge on both temperature and pathogen contamination levels in naturally contaminated raw milk samples, as well as the dynamics of the microbial ecology of raw milk at recommended refrigeration temperatures.

To address the current state of confusion about raw milk microbes and the mathematical relationships describing growth and decline of potential pathogens, RAWMI contracted with Food Safety Net Services, Ltd. (FSNS, San Antonio, TX USA) to conduct a pilot study in properly refrigerated raw milk. FSNS is an independent laboratory certified to quantify the major bacterial pathogens of concern in foods including raw milk (*Campylobacter*, *E. coli* O157:H7, *L. monocytogenes*, and *Salmonella*).

This study provides evidence of microbial growth and decline from a small pilot study on inoculation of raw milk samples with enteropathogens and monitoring during storage for 14 days at 4.4°C (39.9°F), the refrigeration temperature recommended by regulatory agencies in the US. Results of the pilot study are further explored using statistical trend analysis and ANOVA as described herein.

## Materials and methods

### Microbiology methods for FSNS pilot study

Full details on the methodology for the pilot study are provided in the FSNS report [S1 Appendix]. Briefly, inocula were prepared as cocktails of three strains for each of 4 major foodborne bacterial pathogens (*Campylobacter jejuni/coli*, *E. coli* O157:H7, *L. monocytogenes*, *S. enterica* serotypes Enteritidis/Seftenberg/ Typhimurium) as documented by Brandt ([32]; see also Supplementary Materials).

- *Campylobacter jejuni* ATCC 33291 and 33560; *C. coli* ATCC 33559,

- *E. coli* O157:H7, ATCC 700599 and ATCC 43895, food isolates; ATCC 35150, human isolate

- *L. monocytogenes*, ATCC 19115, Serotype 4b and ATCC 7644, Serotype 1/2c, human isolates; ATCC 19114, Serotype 4a, animal isolate

- *S. enterica* serotypes Typhimurium ATCC 14028 and Seftenberg 775W ATCC 43845, food isolates; Enteritidis ATCC 49218)

Duplicate samples of hygienic raw milk produced for direct human consumption by a RAWMI-listed dairy (Raw Farm, formerly Organic Pastures, Fresno, CA USA) were inoculated with one of two initial levels of each of the 4 pathogens (moderate levels ranging from 22 to 162 CFU/mL; high levels ranging from 600 to 8,300 CFU/mL). The inoculated raw milk samples were incubated at 4.4˚C.

Pathogens were quantified over time after inoculation (days 0, 3, 6, 9, 12, and 14) by standard culture-based methods. Aliquots of inoculated raw milk were spread plated on selective agar plates (Campy-Cefex, Xylose Lysine Deoxycholate, Modified Oxford, and Sorbitol Mac-Conkey with Cefixime and Tellurite (CT-SMAC) for the enumeration of *Campylobacter*, *S. enterica*, *L. monocytogenes*, and *E. coli* O157:H7, respectively). Typical colonies were counted from each of the countable plates and recorded as colony forming units per mL (CFU/mL). The experiments were conducted in triplicate, producing a total of 48 time-series observations by pathogen and initial inoculation levels measured as CFU/mL over the 14 days of refrigerated storage.

In addition to the enumeration results for pathogen growth and decline, the pilot study report [32] also documents pH (range 6.3–7.1) and enumeration results for indicator organisms for milk quality (total aerobic plate counts (APC), total LAB, total coliforms, total yeasts, and molds (YM), and psychrotrophs) at days 0 and 14 for uninoculated raw milk samples.

## Statistical methods

The Mann-Kendall Test, a nonparametric statistical test to detect a monotonic trend in time-series data, was performed to detect a statistically significant increasing or decreasing trend in 48 time-series observations generated by FSNS in the pilot study [32]. The Mann-Kendall Test [33] compares each data point to every successive measurement and determines if the change is positive or negative (the magnitude of change, or slope, is not considered). Each discordant pair is given a score of -1 and concordant pairs a score of +1. Tied values are given a score of 0. A test statistic ('S') is then computed based on the difference between the number of positive differences and negative differences. The sign of the S value indicates the overall direction of the data over time but must be compared to a critical value based on a 95% confidence level to accept or reject the null hypothesis of no trend (equal numbers of positive and negative differences).

The Mann-Kendall Test was applied to each pathogen, lot of milk, and technical replicate for each of 48 individual time-series observations. Mann-Kendall calculations were performed in R version 4.1.1 [34] using the Kendall package version 2.2.1 [35]. Significance was assessed at ($\alpha = 0.05$); p values <0.05 were considered significant.

The effect of time (day of storage) on pathogen number ($\log_{10}$ CFU/mL) within a genus and initial level was analyzed by one-way, analysis of variance (ANOVA) using Prism version 9.2 (GraphPad Software Inc., San Diego, CA). When ANOVA was significant ($\alpha = 0.05$) for time, mean pathogen number among days of storage within a pathogen and initial level were compared using Tukey's multiple comparison test at $P < 0.05$. Some samples of milk stored

for 9 to 14 days at 4.4°C tested negative (0 CFU/mL) for *Campylobacter*. Because there is no $\log_{10}$ value for zero, a default value of -0.01 $\log_{10}$ CFU/mL was used for ANOVA. This default value was based on an accepted convention used in ComBase, an international microbial modeling database, for these types of data [36].

## Results

### FSNS pilot study

The pilot study conducted by the contract laboratory FSNS [32] demonstrated that *Campylobacter* spp., *E. coli* O157:H7, and *Salmonella enterica* spp. did not grow in raw milk at 4.4°C during 14 days of refrigerated storage. Similarly, *L. monocytogenes* did not grow at this temperature until day 9 for the lower initial inoculum (26 to 41 CFU/mL) and day 6 for the higher initial inoculum (3,000 to 7,900 CFU/mL), respectively.

The range of initial counts of indicators for the pilot study conducted by FSNS ([32]; full report provided in Supplemental Materials) are provided parenthetically: total aerobic plate counts (510 to 1,900); psychrotrophic plate counts (10–200,000); total coliforms (10–50), total lactic acid bacteria (70–470), and yeasts and molds (10–20). Although some indicators grew and some declined by day 14 [S1 Appendix], no statistical testing for correlations between indicators and pathogens were conducted for the pilot study.

### Mann-Kendall and ANOVA analysis

Results of our statistical analyses of pathogen trends in the pilot study data are presented in Table 1 and Figs 1 and 2. Mann-Kendall statistics are presented in Table 1, plots of the 48 time-series observations by pathogen and initial inoculation level are depicted in Fig 1, and results of ANOVA are presented in Fig 2.

During the first week of refrigerated storage, evidence of pathogen growth was not documented by ANOVA (Fig 2). In the second week of monitoring, evidence of *L. monocytogenes* growth was documented by the Mann-Kendall Test for trend in 8 of 12 replicates (P = 0.004 to P = 0.043; Table 2) and ANOVA (Fig 2). No evidence of trend or significant evidence of decline was observed using the Mann-Kendall Test for *Campylobacter*, *E. coli* O157:H7, and *Salmonella*.

Further, results from ANOVA (Fig 2) indicated that the inoculated pathogen *Campylobacter* declined continuously in milk stored at 4.4°C until it was eliminated from a lower initial level (2.0 $\log_{10}$ CFU/mL) after 9 days of storage (Fig 2A) or from a higher initial level (3.0 $\log_{10}$ CFU/mL) after 12 days of storage (Fig 2B). In contrast, *E. coli* O157:H7 declined initially (days 0 to 3 of storage) but then survived (days 3 to 14 of storage) at the reduced level, resulting in a small to moderate (0.5 to 0.9 $\log_{10}$ CFU/mL) but significant reduction of its initial lower (Fig 2C) or higher (Fig 2D) levels in milk. Similarly, *Salmonella* declined initially (day 0 to 3 of cold storage) and then survived at a reduced level (Fig 2E) or died slowly throughout refrigerated storage (Fig 2F) resulting in a small (0.2 to 0.6 $\log_{10}$ CFU/mL) but significant reduction of its initial levels in milk. The inoculated pathogen *L. monocytogenes* survived initially (days 0 to 6) in milk stored at 4.4°C before it started to grow around day 9 of storage from a lower (1.5 $\log_{10}$ CFU/mL) or higher (3.6 $\log_{10}$ CFU/mL) initial level to a final level of 2.8 (Fig 2G) or 5.3 (Fig 2H) $\log_{10}$ CFU/mL, respectively.

In summary, the ANOVA results indicated that pathogen levels in milk stored at 4.4°C depended on initial level, pathogen genus, and time of storage. Importantly, prolonged storage (9 to 14 days) of milk at 4.4°C significantly reduced or eliminated lower and higher initial levels of *Campylobacter* and resulted in small to moderate (0.2 to 0.9 $\log_{10}$ CFU/mL) but significant reductions in lower and higher initial levels of *E. coli* O157:H7 and *Salmonella*. However,

**Table 1. Results of the Mann-Kendall Test for Trend for 48 time-series observations by pathogen and initial inoculum level.**

| Pathogen | Lot | Replicate | n | Mann-Kendall Test Values | | |
|---|---|---|---|---|---|---|
| | | | | S | p-value | conclusion |
| *Campylobacter* (43 to 162 CFU/mL initial inoculum) | A | 1 | 6 | -10 | **0.035** | ↘ |
| | | 2 | 6 | -12 | **0.013** | ↘ |
| | B | 1 | 6 | -12 | **0.013** | ↘ |
| | | 2 | 6 | -9 | **0.036** | ↘ |
| | C | 1 | 6 | -10 | **0.035** | ↘ |
| | | 2 | 6 | -12 | **0.013** | ↘ |
| *Campylobacter* (600 to 1,100 CFU/mL initial inoculum) | A | 1 | 6 | -12 | **0.013** | ↘ |
| | | 2 | 6 | -12 | **0.013** | ↘ |
| | B | 1 | 6 | -12 | **0.018** | ↘ |
| | | 2 | 6 | -8 | 0.090 | NT |
| | C | 1 | 6 | -12 | **0.018** | ↘ |
| | | 2 | 6 | -14 | **0.006** | ↘ |
| *L. monocytogenes* (26 to 41 CFU/mL initial inoculum) | A | 1 | 6 | 11 | **0.030** | + |
| | | 2 | 6 | 10 | **0.043** | + |
| | B | 1 | 6 | 11 | **0.030** | + |
| | | 2 | 6 | 11 | **0.030** | + |
| | C | 1 | 6 | 9 | 0.066 | NT |
| | | 2 | 6 | 9 | 0.066 | NT |
| *L. monocytogenes* (3,000 to 7,900 CFU/mL initial inoculum) | A | 1 | 6 | 13 | **0.012** | + |
| | | 2 | 6 | 9 | 0.066 | NT |
| | B | 1 | 6 | 13 | **0.012** | + |
| | | 2 | 6 | 15 | **0.004** | + |
| | C | 1 | 6 | 13 | **0.012** | + |
| | | 2 | 6 | 9 | 0.066 | NT |
| *E. coli* O157:H7 (22 to 46 CFU/mL initial inoculum) | A | 1 | 6 | -3 | 0.354 | NT |
| | | 2 | 6 | -1 | 0.500 | NT |
| | B | 1 | 6 | -3 | 0.354 | NT |
| | | 2 | 6 | -3 | 0.354 | NT |
| | C | 1 | 6 | -8 | 0.090 | NT |
| | | 2 | 6 | -4 | 0.283 | NT |
| *E. coli* O157:H7 (6,700 to 8,300 CFU/mL initial inoculum) | A | 1 | 6 | -7 | 0.130 | NT |
| | | 2 | 6 | -5 | 0.226 | NT |
| | B | 1 | 6 | -5 | 0.226 | NT |
| | | 2 | 6 | -1 | 0.500 | NT |
| | C | 1 | 6 | -9 | 0.066 | NT |
| | | 2 | 6 | -3 | 0.354 | NT |
| *Salmonella* (56 to 86 CFU/mL initial inoculum) | A | 1 | 6 | 9 | 0.066 | NT |
| | | 2 | 6 | -3 | 0.354 | NT |
| | B | 1 | 6 | -9 | 0.066 | NT |
| | | 2 | 6 | 1 | 0.500 | NT |
| | C | 1 | 6 | -11 | **0.030** | ↘ |
| | | 2 | 6 | -12 | **0.018** | ↘ |
| *Salmonella* (3,700 to 6,200 CFU/mL initial inoculum) | A | 1 | 6 | -11 | **0.030** | ↘ |
| | | 2 | 6 | -7 | 0.130 | NT |
| | B | 1 | 6 | -12 | **0.018** | ↘ |
| | | 2 | 6 | -15 | **0.004** | ↘ |

*(Continued)*

**Table 1.** (Continued)

| Pathogen | Lot | Replicate | n | Mann-Kendall Test Values | | |
|---|---|---|---|---|---|---|
| | | | | S | p-value | conclusion |
| | C | 1 | 6 | -9 | 0.066 | NT |
| | | 2 | 6 | -15 | **0.004** | ↘ |

Notes: Lot = bottle of raw milk; A, B, and C were bottles from the same day of production; Replicate = duplicate analyses from each lot of raw milk; n = number of observations in each time series; S = Mann-Kendall test statistic; p-values less than 0.05 shown in bold font indicating a statistically significant increasing or decreasing trend conclusion; Down arrows = Evidence of a statistically significant decreasing trend; + = Evidence of a statistically significant increasing trend; NT = no trend observed.

prolonged storage resulted in significant increases (0.5 to 1.6 $\log_{10}$ CFU/mL) in lower and higher initial levels of *L. monocytogenes*.

Over the 14-day study period at 4.4°C, the average pH of uninoculated raw milk decreased (6.96 to 6.88 for one shipment; 7.1 to 6.4 for the other; see S3 Table in S1 Appendix for detail on individual lots). The contract laboratory documented some variability in counts of microbial indicators for milk quality. For the indicator APC, results decreased after 14 days at 4.4°C

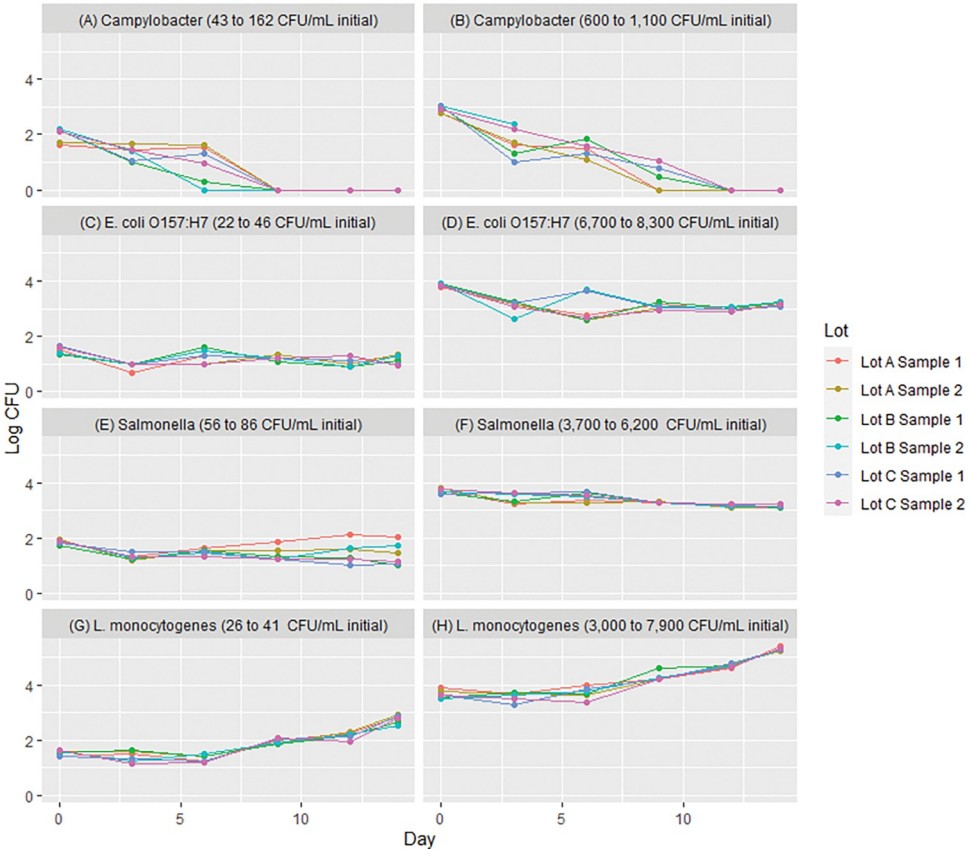

**Fig 1. Plots of the 48 time-series observations by pathogen, initial inoculation levels (labels for each of 8 plots in $\log_{10}$ CFU/mL), and raw milk lots over 14 days of incubation at 4.4°C.** Pathogen counts are reported on the vertical axis in $\log_{10}$ CFU/mL, with time in days post-inoculation on the horizontal axis. Note that 600 CFU/mL in the figure label is equivalent to 2.8 $\log_{10}$ CFU/mL on the vertical axis.

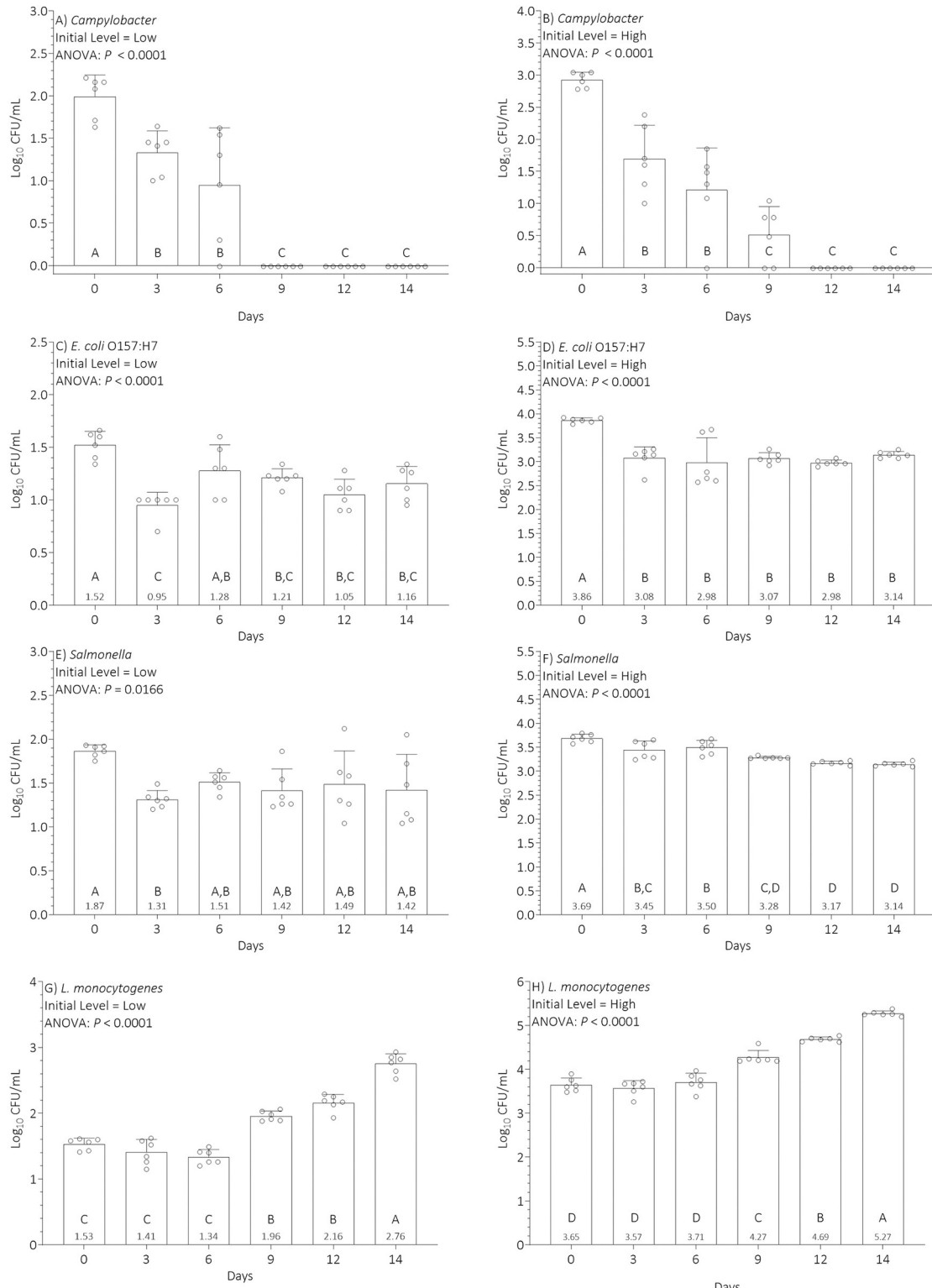

**Fig 2. Analysis of variance for 48 time-series observations by pathogen and initial inoculation level over 14 days of incubation at 4.4˚C.** Pathogen counts are reported on the vertical axis log10 CFU/mL, with time in days post-inoculation on the horizontal axis. Observations with different letters denote statistical significance P ≤ 0.05 using Tukey's multiple comparison test. Note that 600 CFU/mL in the figure label is equivalent to 2.8 $\log_{10}$ CFU/mL on the vertical axis.

**Table 2. Guidelines for microbiological risk assessment [47] highly relevant for this study.**

| Guideline # | Description |
|---|---|
| 1 | Microbiological Risk Assessment should be soundly based upon science. |
| 5 | The conduct of a Microbiological Risk Assessment should be transparent. |
| 9 | A Microbiological Risk Assessment should explicitly consider the dynamics of microbiological growth, survival, and death in foods and the complexity of the interaction (including sequelae) between human and agent following consumption as well as the potential for further spread. |
| 10 | Wherever possible, Risk Estimates should be reassessed over time by comparison with independent human illness data. |
| 11 | A Microbiological Risk Assessment may need reevaluation, as new relevant information becomes available. |

for the first shipment (from 1,643 to 663 CFU/mL) and increased for the second (from 1,020 to 666,000,000 CFU/mL). Similarly, for total LABs, results increased for the first shipment (from 80 to 263) and decreased for the second (from 423 to 10 CFU/mL). For the remaining indicators, results increased for both shipments (total coliforms from 30 to 50 and from 10 to 1,007 CFU/mL; total YM from 10 to 81,667 and from 20 to 7,203 CFU/mL; and psychrotrophs (from 20 to too numerous to count (>57,000,000,000 CFU/mL and from 133,333,333 to >2,500,000,000 CFU/mL).

Note that the pilot study was not designed to perform statistical testing for correlations between indicators and pathogens, nor for fitting parameter values for pathogen growth and decline curves. See S1 Appendix for detail on counts of pathogens and indicators for individual lots.

## Discussion

This small pilot study was undertaken to measure pathogen counts in inoculated samples of raw milk produced for direct human consumption and stored at 4.4˚C for two weeks, to estimate statistical trends for growth and decline, and to challenge misinformation about pathogen growth in raw milk complete with its natural microbiota. Data from the small pilot study was sufficient to estimate trends of pathogen decline in the first week and to conduct ANOVA, but insufficient to estimate parameters of growth and decline for the 48 time series curves for the inoculated pathogens (Table 1 and Figs 1 and 2). The pilot study data and trends are consistent and provide statistically significant results by both the Mann-Kendall Test and ANOVA. More research is needed to enable parameter estimations and deeper statistical characterization of pathogen growth and decline in raw and pasteurized milks for future QMRA simulations.

Temperature, as well as competition with the natural microbiota, are widely recognized as key factors for controlling microbial growth in foods [30], also key for managing raw milk risks as pointed out by the European Food Safety Authority [37]. Dairy farmers and retailers are trained to rapidly cool raw milk and continuously monitor refrigeration temperatures in chill tanks, trucks, and retail refrigeration cases. Consumers are advised to transport refrigerated foods with a cold pack in an insulated bag and keep their refrigerators set at 4.4˚C, and deviations or noncompliance with recommendations can be represented in QMRA abuse scenarios.

The major finding of the pilot study is statistical evidence of no growth at 4.4˚C for the major foodborne pathogens causing illness associated with raw milk in the US (*Campylobacter*, *E. coli* O157:H7, and *Salmonella*; [38]. For listeriosis, rarely associated with illness from raw

milk, the pilot study documented evidence of pathogen growth in 8 of 12 replicates (P = 0.001 to P = 0.028, significant by ANOVA in the second week of refrigerated storage).

An extensive body of evidence [30, 31, 39–41] documents both intrinsic factors (moisture content, pH, nutrient and micronutrient content, biological structure, redox potential, naturally occurring or added antimicrobials, and competitive microbiota) and extrinsic factors (packaging atmospheres, time and temperature effects, storage or holding conditions, and both thermal and non-thermal processing steps) that drive or suppress microbial growth in foods. These studies also document extensive evidence of synergy (or greater benefit) for multiple barriers to pathogen growth or 'hurdles' acting via different cellular mechanisms. Combinations of hurdles (e.g., pH, naturally occurring antimicrobials, refrigeration, and competitive microbiota) can prevent multiplication, inactivate, or kill pathogens in foods while maintaining nutrient content and improving stability, safety, and quality of foods [41]. Suppression of pathogen growth in properly refrigerated raw milk demonstrated herein and in previous studies [7, 8, 10] are consistent with multi-hurdle risk management.

## Need for reliable data to replace invalid assumptions for robust risk analysis

The major limitations of the pilot study are that raw milk from a single US dairy was analyzed and time-series observations at only 3 time points were conducted in the first week, and a total of 6 time points over 14 days of refrigerated storage post-inoculation. However, another published study [42] also documented time series including only 6 time points for refrigerated storage of raw milk. Further, these researchers inoculated extremely high levels of an enteropathogen ($10^5$ or 100,000 cfu/mL) despite detecting 35 or fewer pathogens per mL from naturally contaminated milk from the same dairy (range 0.007 to 35 MPN/mL; [42]. It is unclear if the reported trends from the extremely high inoculated levels would be consistent with trends for raw milk samples inoculated at levels 4 or more orders of magnitude lower. In addition, Jaakkonen and colleagues did not obtain fresh raw milk from the producer for their study on survival trends, but reported purchasing raw milk in the retail market. It is uncertain if results reported herein and by Jaakkonen and colleagues [42] are representative of other conditions, particularly due to documentation of high variability of the raw milk microbiota across herds, farms, breeds, diets, storage times and temperatures, and seasonality [43–45].

Further research is needed to quantitate pathogen growth and decline rates for raw milk inoculated at levels of pathogen contamination observed in fresh naturally contaminated samples in order to minimize bias and optimize the experimental design to reflect feasible ecological conditions for predictions in a complex food. Unbiased data are essential for assessing and re-assessing risks for raw and pasteurized milks from multiple dairy farms.

This data gap for predictive microbiology of pathogens in raw and pasteurized milks is relevant to microbial risk assessment because two historic QMRAs [5, 46] appeared to select intentionally conservative assumptions. Both QMRAs appear subject to overestimation bias for raw milk risks. Neither QMRA included or discussed data demonstrating that pathogen growth is slower in raw milk than pasteurized milk, attributable in part to pathogen competition with the dense and diverse natural microbiota of milks. Despite characterizing both raw and pasteurized milks as high-risk foods for severe listeriosis, FDA/FSIS [5] selected different risk communications and risk management policies that were not based on the scientific evidence. The risk management policies for prohibition and recommended avoidance, respectively, for raw milk in Australia [46] and the US [5] are inconsistent with both then available and current scientific evidence discussed herein, and in more detail by [2].

## Need for evaluation of QMRAs relative to international guidance and quality criteria

The pilot study design was also motivated by the consensus statement on general principles and guidelines for QMRA ratified by 163 member countries of the Codex Alimentarius Commission (CAC) in 1999 [47]. Of the 11 CAC principles, five are highly relevant to this study (Table 2).

Considered together, these principles focus on sound and transparent processes, including use of the best available scientific evidence for modeling microbiology ecology, as well as reassessing and reevaluating over time as science advances. In addition, these principles acknowledge that risk assessors may choose to apply assumptions rather than scientific data when significant gaps in knowledge exist.

In order to fully address these principles, risk practitioners relying on assumptions rather than objective scientific data must also characterize the implications of alternative assumptions and their impact on risk estimates and scenarios for risk management options, critical aspects of quality risk analysis, as articulated in the Risk Analysis Quality Test (RAQT) of the Society for Risk Analysis (SRA; S2 Appendix, also available at https://www.sra.org/risk-analysis-specialty-groups/applied-risk-management/scientific-literature/). Both historic government QMRAs [5, 46] failed most or all of the questions for evaluating risk analysis quality that frame the 76-question battery of the RAQT (workshop manuscript in development through the SRA Applied Risk Management specialty group).

Significant gaps in knowledge for raw milk QMRAs were raised as needs for future reassessment in two historic QMRAs that examined raw milk [5, 46], as well as in a more recent review that included two FDA contributors to the assessment [27]. These gaps remain unfilled to date. The FDA/FSIS risk assessment team inappropriately assumed that growth rates for *L. monocytogenes* were equivalent for pasteurized and raw milks, despite data to the contrary. Some year later, the Food Standards Australia New Zealand team conducted a QMRA for raw cow milk [46] that was largely based on unvalidated assumptions and extrapolations rather than reliable data for raw milk. Thus, both QMRAs imposed overestimation bias on their assessments for raw milk and did not fully disclose the impacts of their intentionally conservative assumptions on risk estimates, management options for risk reduction, or risk communications.

Regarding guideline 10, the US Centers for Disease Control and Prevention provided a recent dataset for outbreaks from all transmission sources including both raw and pasteurized fluid milks for the period 2005 to 2020 [38]. Raw and pasteurized milks both caused nearly 2,000 illnesses over this 16-year period [38] (manuscript in preparation). Mortality rates associated with milks in North America in recent decades are quite low, including 5 US fatalities (3 associated with pasteurized milk and 2 with raw milk; [38], and 4 Canadian fatalities associated with pasteurized milk [48].

Perhaps the most highly relevant general principle for QMRAs in this context is guideline 11. The findings of the pilot study, the lack of growth of the foodborne pathogens in raw milk for 14 days at 4.4°C for the major foodborne pathogens causing raw milk outbreaks in the US, are consistent with other peer reviewed studies conducted between 4 and 5°C [7, 8, 10] that falsify the incorrect assumptions about pathogen growth in historic QMRAs.

Two independent academic research teams [9, 11] re-evaluated and extended portions of the historic FDA/FSIS risk assessment for severe listeriosis [5]. Latorre and colleagues [9] estimated risks per raw milk serving to the general population were as low as $10^{-15}$ (~1 illness per 1,000,000,000,000,000 servings), substantially lower estimated risks compared to the FDA/FSIS 2003 assessment that pooled growth data for raw and pasteurized milks. Stasiewicz and

colleagues [11] found in re-assessment that increasing heat treatments increased the growth rates of *L. monocytogenes* in pasteurized and ultra-pasteurized milks, consistent with killing more of the milk microbiota and thus reducing competition with the pathogen. These researchers also provided supplemental information for their study reporting no growth of the pathogen in raw milk blanks and increasing rates of growth in the raw milk pasteurized for 25 seconds at 72˚ and 82˚C. The need to update incorrect assumptions and misinformation about both predictive microbiology and dose-response relationships in this QMRA was raised for future re-assessment for the FDA/FSIS QMRA [49].

## Need for transparency about scientific evidence falsifying prior assumptions

The need to update incorrect assumptions about pathogen growth in raw milk that were made in historical QMRAs [5, 46] is more urgent than ever because so many claims about raw and pasteurized milks are made in the media, as well as in the scientific literature, without rigorous supporting data. Consumers and scientists can understandably be confused by conflicting claims. SRA leaders seek to encourage others to apply the SRA RAQT in both review of completed QMRAs for other foods and water, as well as in planning for future risk analysis projects, with the goal of developing a culture of full disclosure and quality analysis. Cultural, social, or ideological constructions have in the past limited the influence of scientific evidence into policy making, as documented by Meagher and colleagues [50] on cultural mischaracterization of two foodborne outbreaks. Despite quick tracing of a 2006 outbreak to California-grown spinach, FDA's public risk communication to avoid consuming any raw spinach contributed to market collapse, and "a range of plausible responses were never considered" ([50], pg. 245). Similarly, organizations around the world appear to incorrectly attribute high risk to raw milk from all producers, and not to any source of pasteurized milk. Participants in the SRA workshop on risk analysis quality discussed common unstated and unsupported assumptions about milks include: 1) the source of microbes in milk is feces; 2) raw milks are inherently dangerous; 3) pasteurization is a 'silver bullet'; and 4) pasteurized milk is zero risk (manuscript in preparation). Current evidence documented herein and by Coleman and colleagues [15] and Dietert and colleagues [2] supports none of these assumptions.

The importance of correctly modeling the microbial ecology of raw milk demonstrated by LAB strains isolated from raw bovine milk suppression or exclusion of three pathogens inoculated at two high densities, $10^3$ and $10^6$ $\log_{10}$ CFU/mL [19]. Clearly, the natural milk microbiota can suppress the growth of pathogens under some conditions.

From our perspective of available data and analysis consistent with principles of microbial ecology and those of the [47], as well as the RAQT of the SRA, evidence that raw milk is 'inherently dangerous' is lacking. Evidence is consistent with protective multi-hurdle synergies of raw milk including the dense and diverse natural microbiota of mammalian milks under proper refrigeration contributing to suppression or exclusion of pathogens. We also acknowledge that no food is risk free, and benefits and risks could and perhaps should be characterized for all foods.

Further improvements in the credibility and utility of QMRAs might develop with deeper consideration of environmental sustainability, economics and food waste, supply chain structure, climate change, and social and cultural factors [51–53]. For example, Duret and colleagues [51] determined that setting the domestic refrigerator temperature to 4˚C presented the best compromise for balancing risk of foodborne illness, food waste, and energy consumption. Rendueles and colleagues [31] suggest not only that multi-hurdle approaches can reduce risk of illness and maintain food quality, but also can support more sustainable food production chains in the global market.

Thus, for design of future predictive microbiology studies to inform risk analysis, studies must include additional production lots, dairy farms, and regions or states to characterize regional or national trends for milk risks. Expansions of the pilot study design should also include more frequent sampling and multiple initial inoculation levels for pathogens (at least ~1 CFU/mL and ~1,000 CFU/mL) so that robust parameters for growth and decline can be estimated. An ideal study design might also explore potential mechanisms of pathogen suppression under proper refrigeration and temperature abuse scenarios by quantitating key representatives of the raw milk microbiota over the study period and identifying microbial associations that drive pathogen suppression and killing. Rigorous quantitative data on predictive microbiology of raw milks is essential to re-evaluating historic QMRAs based on invalid assumptions about pathogen growth in raw milk so that unbiased estimates of risks and benefits can be generated for raw and pasteurized milks.

## Conclusion

Results from a small pilot study with fresh raw milk produced for direct human consumption were consistent with previous studies demonstrating suppression of growth of major bacterial pathogens at proper refrigeration temperatures. Future research is needed to expand the results of the small pilot study on pathogen suppression in raw milks to address risk analysis more holistically, structuring and simulating tradeoffs between benefits and risks of raw and pasteurized milks.

## Supporting information

**S1 Appendix. FSNS report [32], determination of growth rate of *Salmonella enterica* spp., *E. coli* O157:H7, *Campylobacter* spp., and *Listeria monocytogenes* in raw milk.**
(PDF)

**S2 Appendix. Risk analysis quality test of the society for risk analysis.**
(PDF)

**S3 Appendix. Dataset from pilot study.**
(XLSX)

**S1 Table. Published studies on pathogen growth and decline in raw milk at refrigeration and abuse temperatures.**
(DOCX)

## Author Contributions

**Conceptualization:** M. E. Coleman, T. P. Oscar, T. L. Negley.

**Data curation:** M. E. Coleman, T. P. Oscar, T. L. Negley.

**Formal analysis:** T. P. Oscar, T. L. Negley.

**Funding acquisition:** M. E. Coleman.

**Methodology:** M. E. Coleman, T. P. Oscar, T. L. Negley.

**Supervision:** M. E. Coleman.

**Visualization:** M. M. Stephenson.

**Writing – original draft:** M. E. Coleman, T. P. Oscar.

**Writing – review & editing:** T. P. Oscar, T. L. Negley, M. M. Stephenson.

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
