## [Decision Letter · Decision Letter 0]

26 Apr 2023

PONE-D-23-04758Suppression of Pathogens in Properly Refrigerated Raw MilkPLOS ONE

Dear Dr. Oscar,

Thank you for submitting your manuscript to PLOS ONE. After careful consideration, we feel that it has merit but does not fully meet PLOS ONE’s publication criteria as it currently stands. Therefore, we invite you to submit a revised version of the manuscript that addresses the points raised during the review process.

 Please submit your revised manuscript by Jun 10 2023 11:59PM. If you will need more time than this to complete your revisions, please reply to this message or contact the journal office at plosone@plos.org. Please include the following items when submitting your revised manuscript:A rebuttal letter that responds to each point raised by the academic editor and reviewer(s). You should upload this letter as a separate file labeled 'Response to Reviewers'.A marked-up copy of your manuscript that highlights changes made to the original version. You should upload this as a separate file labeled 'Revised Manuscript with Track Changes'.An unmarked version of your revised paper without tracked changes. You should upload this as a separate file labeled 'Manuscript'.

We look forward to receiving your revised manuscript.

Kind regards,

Arun K. Bhunia, Ph.D.

Academic Editor

PLOS ONE

“M.E.C. serves as an unpaid advisor on the Raw Milk Institute advisory board and has provided expert testimony in several court cases regarding the microbial ecology and assessment of benefits and risks for microbial pathogens in raw milk and other foods and feeds. Other authors have no competing interests to declare.”

Please include your updated Competing Interests statement in your cover letter; we will change the online submission form on your behalf."

Additional Editor Comments:

Review comments are generally favorable; however, reviewer 1 raised concerns about the study design and the wordiness of the discussion section. Discussion section needs to be shortened.

Reviewers' comments:

Reviewer's Responses to Questions

**Comments to the Author**

1. Is the manuscript technically sound, and do the data support the conclusions?

Reviewer #1: Partly

Reviewer #2: Yes

2. Has the statistical analysis been performed appropriately and rigorously? 

Reviewer #1: Yes

Reviewer #2: Yes

3. Have the authors made all data underlying the findings in their manuscript fully available?

Reviewer #1: Yes

Reviewer #2: Yes

4. Is the manuscript presented in an intelligible fashion and written in standard English?

Reviewer #1: Yes

Reviewer #2: Yes

5. Review Comments to the Author

Reviewer #1: The author studied the growth curves of several pathogenic bacteria in raw milk for direct drinking, and compared them with relevant public reports, providing some support for the change of the growth kinetics of pathogenic bacteria in raw milk. However, there are some defects in the experimental design, such as the number of pathogenic bacteria accessed is only two levels of control, the sampling interval in the early stage of the test is too wide, and the sampling test is only conducted on the 0, 3, and 6 days, which may have a significant impact on the test results. It is not a new discovery that L.monocytogenes will continue to grow during low-temperature storage. In addition, in the discussion part, the author questioned the role of risk assessment in the formulation of regulations and standards with a large space, which has certain limitations. The test results can not directly prove that the risk assessment has been out of control. In this experiment, Salmonella has been kept at a high level. Although Campylobacter has been inactivated at the end of the shelf life, it is not excluded that consumers can contact products that pollute high-level Campylobacter at the early stage of the shelf life, It is not allowed to exaggerate the safety of direct edible raw milk and belittle the role of pasteurization in milk risk prevention and control without direct evidence. Before considering acceptance, the author should reorganize the discussion section and delete a lot of speculative language.

Reviewer #2: Review of manuscript PONE-D-23-04758 “Suppression of Pathogens in Properly Refrigerated Raw Milk”

Summary

The authors address the issue of conflicts in the literature regarding the growth of Foodborne pathogens in raw milk (Campylobacter, E. coli O157:H7, Listeria monocytogenes, or Salmonella). The introduction provides a review of recent findings concerning both foodborne illnesses incidence associated with the above the fore mentioned bacteria from raw and heat treated milk. They also discuss reports in the literature referring to pathogens inoculated in raw milk having slower growth rates in raw milk than pasteurized milk due to the competing organisms and biologically active components present. The authors commissioned a pilot study through a contract lab to provide further information on the previously mentioned pathogens growth in raw milk.

General Comments

The manuscript is well written.

The pilot study was well designed and used relevant pathogen inoculation numbers.

The statistics were thorough and the data robust.

Specific Comments

The authors make valid points about existing Quantitative Microbial Risk Assessments (QMRA) and their basis on sometimes invalid assumptions. The example from the literature of Listeria growth rates in raw milk being significantly lower than pasteurized milk for instance clearly shows the assumption of similar growth rates for Listeria in raw milk and pasteurized milk to be questionable if not invalid. The pilot study reported in this manuscript addresses the issues associated with previous assumptions and provides compelling support for this argument. I believe this paper should be published because there is a definite need for updating and reassessing current QMRAs as more scientific information becomes available as the authors suggest.

Suggestion

Although this work is concerned with raw milk the authors might want to mention that there are other QMRAs associated with other foods which may need to subjected to this level of scrutiny and there also needs to be some sort of formal mechanism for QMRA reevaluation. A system is in place for the Pasteurized Milk Ordinance (PMO) which changes can be made by proposal submissions at the biennial National Conference of Interstate Milk Shippers.

6. PLOS authors have the option to publish the peer review history of their article (what does this mean?). If published, this will include your full peer review and any attached files.

Reviewer #1: No

Reviewer #2: No

---

## [Author Response · Author response to Decision Letter 0]

5 Jun 2023

A file is included with this submission that responds to the reviewers comments.

---

## [Decision Letter · Decision Letter 1]

14 Jul 2023

Suppression of Pathogens in Properly Refrigerated Raw Milk

PONE-D-23-04758R1

Dear Dr. Oscar,

We’re pleased to inform you that your manuscript has been judged scientifically suitable for publication and will be formally accepted for publication once it meets all outstanding technical requirements.

Kind regards,

Arun K. Bhunia, Ph.D.

Academic Editor

PLOS ONE

Additional Editor Comments (optional):

Accept

Reviewers' comments:

Reviewer's Responses to Questions

**Comments to the Author**

1. If the authors have adequately addressed your comments raised in a previous round of review and you feel that this manuscript is now acceptable for publication, you may indicate that here to bypass the “Comments to the Author” section, enter your conflict of interest statement in the “Confidential to Editor” section, and submit your "Accept" recommendation.

Reviewer #2: All comments have been addressed

2. Is the manuscript technically sound, and do the data support the conclusions?

Reviewer #2: Yes

3. Has the statistical analysis been performed appropriately and rigorously? 

Reviewer #2: Yes

4. Have the authors made all data underlying the findings in their manuscript fully available?

Reviewer #2: Yes

5. Is the manuscript presented in an intelligible fashion and written in standard English?

Reviewer #2: Yes

6. Review Comments to the Author

Reviewer #2: (No Response)

7. PLOS authors have the option to publish the peer review history of their article (what does this mean?). If published, this will include your full peer review and any attached files.

Reviewer #2: No

---

## [Editor Report · Acceptance letter]

27 Jul 2023

PONE-D-23-04758R1 

Suppression of Pathogens in Properly Refrigerated Raw Milk 

Dear Dr. Oscar:

I'm pleased to inform you that your manuscript has been deemed suitable for publication in PLOS ONE. Congratulations! Your manuscript is now with our production department. 

Kind regards, 

on behalf of

Dr. Arun K. Bhunia 

Academic Editor

PLOS ONE